



# Temperature response measurements from eucalypts give insight into the impact of Australian isoprene emissions on air quality in 2050

Kathryn M. Emmerson[1], Malcolm Possell[2], Michael J. Aspinwall[3,4], Sebastian Pfautsch[3], and Mark G. Tjoelker[3]

[1]Climate Science Centre, CSIRO Oceans and Atmosphere, Aspendale, VIC 3195 Australia
[2]School of Life and Environmental Sciences, University of Sydney, Sydney, NSW, Australia
[3]Hawkesbury Institute for the Environment, Western Sydney University, Penrith, NSW, Australia
[4]Department of Biology, University of North Florida, Jacksonville, Florida, 32224 USA

**Correspondence:** Kathryn Emmerson (kathryn.emmerson@csiro.au)

**Abstract.** Predicting future air quality in Australian cities dominated by eucalypt emissions requires an understanding of their emission potentials in a warmer climate. Here we measure the temperature response in isoprene emissions from saplings of four different *Eucalyptus* species grown under current and future average summertime temperature conditions. The future conditions represent a 2050 climate under Representative Concentration Pathway 8.5, with average daytime temperatures of 294.5 K. Ramping the temperature from 293 K to 328 K resulted in these eucalypts emitting isoprene at temperatures 4-9 K higher than default maximum emission temperature in the Model of Emissions of Gases and Aerosols from Nature (MEGAN). New basal emission rate measurements were obtained at the standard conditions of 303 K leaf temperature and 1000 $\mu$mol m$^{-2}$ s$^{-1}$ photosynthetically active radiation and converted into landscape emission factors. We applied the eucalypt temperature responses and emission factors to Australian trees within MEGAN and ran the CSIRO Chemical Transport Model for three summertime campaigns in Australia. Compared to the default model, the new temperature responses resulted in less isoprene emission in the morning and more during hot afternoons, improving the statistical fit of modelled to observed ambient isoprene. Compared to current conditions, an additional 2 ppb of isoprene is predicted in 2050 causing hourly increases up to 21 ppb of ozone and 24-hourly increases of 0.4 $\mu$g m$^{-3}$ of aerosol in Sydney. This forecasted increase in ozone is one fifth of the hourly Australian air quality limit and suggests anthropogenic NO$_X$ should be further reduced to maintain healthy air quality in future.

## 1 Introduction

Biogenic volatile organic compounds (BVOCs) are emitted by vegetation in response to external stressors such as heat, light and herbivory (Sharkey and Monson, 2017). There are hundreds of individual BVOCs all exhibiting different emission behaviours





(e.g. with or without a light dependence), but the largest global flux of a single BVOC is isoprene (2-methyl-1,3-butadiene; $C_5H_8$), with an estimated 440 - 600 Tg C per year (Guenther et al., 2012). Isoprene reacts rapidly in the atmosphere, contributing to ozone ($O_3$) and secondary organic aerosol (SOA) formation. For cities surrounded by forests, BVOC emissions can dominate airsheds contributing to peak summertime ozone (Utembe et al., 2018), and early morning ozone spikes (Millet et al., 2016) if not quenched by the hydroxyl radical (OH) on the previous day.

In addition to the different environmental reasons a plant will emit BVOCs, plants emit their own unique signature of BVOCs with varying strengths, even amongst plants in the same genus. Native to Australia, eucalypt trees are amongst the highest BVOC emitters of any plant species (Benjamin et al., 1996), emitting isoprene constitutively and storing monoterpenes within oil reservoirs in the leaves (Brophy et al., 1991). However, very few of the 800 species in the *Eucalyptus* genus (Department of Agriculture and Water Resources, 2018) have been studied for emissions. This is problematic as biogenic modellers tend to

base simulations on a few measurements which represent a fraction of the potential diversity of species and emission rates. For example, the *Eucalyptus* isoprene emission factors for the Model of Emissions of Gases and Aerosols from Nature (MEGAN) were based on six studies, only one of which was conducted in Australia (see Emmerson et al. (2016)). The Australian study measured large differences of 63 $\mu$g g$^{-1}$ h$^{-1}$ of isoprene between the lowest and highest emitting eucalypt species, with *E. globulus* showing the greatest emission rates (He et al., 2000). Natural occurrence of *E. globulus* is restricted to temperate

south east Australia (including Tasmania).

Use of landscape emission factors (LEF) weighted by higher emitting trees have caused overpredictions in modelled isoprene (Emmerson et al., 2016, 2019a). As young leaves tend to emit more isoprene than older leaves, conducting emission measurements on saplings has been questioned (Street et al., 1997), although adult trees will contain a mixture of leaf ages. However, BVOC emission models such as MEGAN require isoprene emission rates to be determined at standard conditions

of 303 K and 1000 $\mu$mol m$^{-2}$ s$^{-1}$ photosynthetically active radiation (PAR) (Guenther et al., 2012). Measurements made at other temperatures and PAR fluxes need scaling to these standard conditions, which can introduce uncertainties of up to 20 % (He et al., 2000). The standard temperature and light level conditions are better provided for in a controlled greenhouse environment, which necessitates using saplings.

MEGAN describes the emission of BVOCs in terms of temperature, PAR, leaf area index, leaf age, soil moisture, and

suppression via ambient $CO_2$ concentrations. The MEGAN parameterisations are rightly based on a wide range of ecosystem responses to environmental conditions. Many studies have investigated impacts of climate change on isoprene by changing the inputs to MEGAN such as ambient temperatures and $CO_2$ concentrations (e.g. Bauwens et al. (2018), and how land use might change the geographical extent of plant functional types (PFTs) (e.g. Arneth et al. (2011)), without changing the MEGAN parameterisations themselves. Here we report new controlled isoprene response measurements from four eucalypt tree species,

which show different temperature responses than assumed by MEGAN. We also use the controlled experimental conditions to impose a projected 2050 climate to investigate whether eucalypts growing in a warmer climate show a different temperature sensitivity of isoprene emissions than eucalypts growing in the current climate. Accounting for climate warming impacts on isoprene emission capacity provides a lens to study how air quality in Australia could be impacted in the future. Using a





regional chemical transport model allows us to alter the dynamics of MEGAN to suit these new temperature responses for

Australia.

This study aims to i) determine the temperature response of isoprene in four Eucalyptus species grown under two treatments representing current average summertime temperatures and a 2050 climate, and ii) use these measurements to determine the impacts of isoprene in a future climate on predicted levels of $O_3$ and SOA.

## 2   Methods

### 2.1   The MEGAN default temperature response

Guenther et al. (2012) defines the emission of BVOCs in terms of activity factors representing the environmental conditions described above. Here we are interested in studying the temperature response of isoprene, $\gamma T$ (unitless):

$$\gamma T = E_{\text{opt}} \times \left[ \frac{C_{\text{T2}} \times exp(C_{\text{T1}} \times x)}{(C_{\text{T2}} - C_{\text{T1}} \times (1 - exp(C_{\text{T2}} \times x)))} \right] \tag{1}$$

where $E_{\text{opt}}$ is the optimum emission point, and $C_{\text{T1}}$ (95 kJ mol$^{-1}$) and $C_{\text{T2}}$ (230 kJ mol$^{-1}$) are coefficients that fit the response

to a range of ecosystems.

$$x = \left[ \frac{(1/T_{\text{opt}}) - (1/T)}{0.00831} \right] \tag{2}$$

where T is the temperature of the leaf (K) and 0.00831 is the gas constant in kJ K$^{-1}$ mol$^{-1}$. The optimum temperature for emission in MEGAN, $T_{\text{opt}}$ is calculated below.

$$T_{\text{opt}} = T_{\text{max}} + (0.6 \times (T_{240} - T_{\text{S}})) \tag{3}$$

$$E_{\text{opt}} = C_{\text{eo}} \times exp(0.05 \times (T_{24} - T_{\text{S}})) \times exp(0.05 \times (T_{240} - T_{\text{S}})) \tag{4}$$

where $T_{\text{max}}$ is 313 K, $T_s$ is the standard leaf temperature (297 K), and $T_{24}$ and $T_{240}$ are the average leaf temperatures of the previous 24 and 240 hours, respectively. $C_{\text{eo}}$ is an empirical coefficient of 2 for isoprene.

### 2.2   Experimental conditions

Four eucalypt species were chosen based on their prevalence in Australia, and in particular New South Wales (Table 1). Two

of the trees, *E. camaldulensis* and *E. tereticornis* have a wide geographical representation within Australia, having a latitudinal native growing range of 9-38 °S (Atlas of Living Australia, 2019). The native climatic distribution range of the other species, *E. botryoides* and *E. smithii* are restricted to the south east coastal regions. We will use our new experimental data to revise the LEF maps for Australia, weighting the results according to the summed area of the four species (Table 1).

Plant species can be classified as low (less than 1 $\mu$g g$^{-1}$ h$^{-1}$), moderate (1-10 $\mu$g g$^{-1}$ h$^{-1}$) or high (greater than 10 $\mu$g g$^{-1}$ h$^{-1}$)

isoprene emitters (Benjamin et al., 1996). Of the four eucalypts used in this study, *E. camaldulensis* and *E. tereticornis* are high isoprene emitters (Table 1), whilst *E. botryoides* is classed as moderate. The emission category of *E. smithii* is unknown. All tabulated measurements were scaled to the standard conditions from other temperatures and PAR.





Eighty trees (20 of each species) were grown from seed at Hawkesbury Institute for the Environment in Richmond, NSW. After eight weeks seedlings were transplanted into 6.9 L pots filled with alluvial soil and split randomly into two treatment groups, each containing 10 seedlings of each species. The first treatment group was grown for 85 days at an average daily temperature of 291 K (current climate) and the second treatment group was grown for 85 days at 294.5 K (future climate). In this time the seedlings put on vigorous growth and developed into ∼1.5 m tall saplings with plenty of leaves (see photograph in supplementary). The future climate treatment represents temperature conditions in Australia in 2050 assuming the highest 8.5 Representative Concentration Pathway (RCP) – the business as usual scenario where $CO_2$ reaches 940 ppm by 2100 (van Vuuren et al., 2011). The treatments maintained the diurnal variation of ambient temperature at 9 K. Further details on the growth conditions of these eucalypts are described in Aspinwall et al. (2019), prior to their study of how eucalypts respond to heatwave stress.

The measurements conducted on the future climate-grown trees provides the opportunity to study how isoprene emissions could change across Australia in a 2050 climate. This opportunity assumes the four eucalypt species exist in a 2050 climate and continue to have Australian coverage. Forward modelling suggests both *E. camaldulensis* and *E. tereticornis* will grow in Australia in 2085 (González-Orozco et al., 2016).

## 2.3 Temperature response measurements

Leaf gas exchange measurements were made with a LI-6400XT portable photosynthesis system (Li-Cor Inc., Lincoln, NE, USA) connected to a Walz 3010-GWK1 leaf cuvette (Heinz Walz GmbH, Effeltrich, Germany). The Walz cuvette was con- trolled via a PC using Walz software (GFS-Win v.3.47g). $CO_2$ concentrations were set to 400 ppmv and the flow rate through the cuvette was set to 700 $\mu$mol s$^{-1}$. Light was provided using Lumigrow Pro 325 LED growth lamps (LumiGrow, Novato, CA) positioned above the cuvette to provide 1000 $\mu$mol m$^{-2}$ s$^{-1}$ PAR as measured by the LI-6400XT cuvette's light sensor. Leaf temperature was controlled using the Walz cuvette and was programmed to increase leaf temperature in 5 K steps from 293 K to 328 K in seven minute intervals to accommodate adjustment to new steady state values of photosynthesis at each temperature. This time corresponds to the duration of intermediate length sunflecks in plant canopies (Pearcy, 1990) and also results in a common, standardised heat dose for all the leaves (Niinemets and Sun, 2015). Basal emission rates are taken as the emission rate measured at 1000 $\mu$mol m$^{-2}$ s$^{-1}$ PAR and 303 K.

After the gas exchange measurements, leaves were detached and their area measured using a LI-3100C leaf area meter (Li-Cor Inc.). Leaves were oven dried at 105 °C for 72 hours after which their dry weight was recorded.

Mixing ratios of isoprene by volume were determined using a high-resolution proton transfer reaction-mass spectrometer (PTR-MS, Ionicon GmbH, Innsbruck, Austria). The operating parameters of the PTR-MS were held constant during mea- surements, except for the secondary electron multiplier voltage, which was optimised before every calibration. The drift tube pressure, temperature and voltage were 2.2 hPa, 50 °C and 600 V. The parameter E/N was ∼125 Td ($1.25 \times 10^{-15}$ V cm$^2$) and the reaction time was ∼100 $\mu$s. The count rate of $H_3O^+ \cdot H_2O$ ions was 1–2 % of the count rate of $H_3O^+$ ions, which was $5.0 - 5.5 \times 10^6$ s$^{-1}$. Normalized sensitivities and isoprene volume mixing ratios were calculated through calibrations as described by Taipale et al. (2008) using 5 ppmv isoprene (Apel-Riemer Environmental Inc., Broomfield, CO) diluted in high





purity nitrogen (BOC Ltd, Sydney, NSW). Protonated isoprene was detected by the PTR-MS as its molecular mass plus one (i.e. M + H+1 = 69). The duty cycle for each measurement period was 5 s.

Isoprene-temperature response measurements were replicated on five or six saplings of each species in each temperature
treatment group. The Solver program (Generalized Reduced Gradient nonlinear method, default settings; Microsoft Excel for Office 365; Microsoft Corporation, Redmond, WA, USA) was used to estimate four MEGAN coefficients, $C_{T1}$, $C_{T2}$, $T_{max}$ and $C_{eo}$ to minimise the difference between the result of Equation 1 and the measured temperature responses, for each tree species and growth temperature treatment. The basal emission rates for each species (in $\mu g\ g^{-1}\ h^{-1}$) were normalised to the average basal emission factor for that species and its growth temperature treatment. Normalising these data scales the actual emission
rates and ensures they have a common basal emission factor of unity.

## 2.4 Observations of isoprene mixing ratios

Few measurements of ambient isoprene exist in Australia. Hourly observations made by Proton Transfer Reaction Mass Spectrometry are available for three summertime urban field campaigns near Sydney (Figure 1). These observations will be used to evaluate model predictions using our temperature response functions of isoprene emission. Isoprene observations are avail-
able from Bringelly in the January-February of 2007, SPS1 in Westmead in the February-March of 2011 (Keywood et al., 2019), and MUMBA in Wollongong in January-February of 2013 (Paton-Walsh et al., 2017, 2018). Climate projections for Australia forecast increases in average temperatures with an accompanying increase in the frequency of extreme heatwave days (Bureau of Meteorology and CSIRO, 2018). There were several hours across these campaigns where maximum temperatures exceed 303 K and 313 K. Temperatures above 303 K occurred for 37 hours at Bringelly, 29 hours in SPS1, and 27 hours during
MUMBA. There were six hours during MUMBA where temperatures exceeded 313 K, with a maximum of 317.2 K on January 18th 2013.

## 2.5 The CSIRO Chemical Transport Model (C-CTM)

The C-CTM is a modelling framework designed to predict the atmospheric concentrations of gases and aerosols due to emissions, transport, chemical production and loss, and deposition. In addition to BVOCs, the framework has successfully predicted
pollen (Emmerson et al., 2019b), health effects from shipping (Broome et al., 2016) and air quality (Chambers et al., 2019). The C-CTM is driven by meteorology from the Conformal Cubic Atmospheric Model (CCAM, McGregor and Dix (2008)), taking boundary conditions from ERA-Interim. Four nested domains are used at spatial resolutions of 80 km, 27 km, 9 km and 3 km to downscale the atmospheric constituents over topography that increases with complexity at higher resolutions. The inner 3km domain contains 114 x 110 gridcells to encompass Sydney, Wollongong and the surrounding forested regions
(Figure 1).

The model chemistry scheme is MOZART-T1 (Emmons et al., 2020) incorporating the latest research on isoprene oxidation pathways via additional radical production under low $NO_X$ conditions. The aerosol framework is a two-bin sectional scheme, processing organic species by the Volatility Basis Set (Shrivastava et al., 2008) and processing inorganic species via



ISORROPIA_II (Fountoukis and Nenes, 2007). The high and low $NO_X$ aerosol mass yields for the organic species, including
isoprene, are provided by Tsimpidi et al. (2010).

Australia wide anthropogenic emissions come from an inventory based on human population density on a 10 km x 10 km grid
resolution (updated from Physick et al. (2002)). Anthropogenic emissions for Sydney in the 3km domain are based on the most
recent NSW inventory for the year 2008 (EPA NSW, 2012). The full canopy environment version of MEGAN2.1 (Guenther
et al., 2012) was built into the C-CTM to calculate the biogenic emissions (Emmerson et al., 2016). Isoprene emissions, R in a
given grid cell, xy, are predicted using LEF maps in combination with the land fraction, $\chi$ occupied by 16 PFTs, j, using:

$$R = LEF_{x,y} \sum_{j=1}^{nPFT} (\gamma_{x,y} \times \chi_j) \tag{5}$$

Where $\gamma$ represents the sum of all activity factors for light, temperature, soil moisture, leaf area index and leaf age. The $\gamma$ for
soil moisture is applied using data provided by the Soil-Litter-Iso model (SLI), as recommended by Emmerson et al. (2019a).
Monthly leaf area index data come from MODIS MCD15A2 version 4.

A PFT map based on the ESA CCI Land Cover distribution for the year 2010 (ESA, 2016) was created. The ESA land-
cover data was used in conjunction with MODIS 44B (Vegetation Continuous Fields) product, level 5.1 for the year 2012 to
provide the percentage tree, grass and shrub cover. Details on how these landcover data were aggregated or split into the 16
PFTs required by MEGAN2.1 are provided in the supplementary. Eucalypts fall under the broadleaf evergreen temperate tree
category.

# 3   Results and discussion

## 3.1   Temperature response results

The fitted temperature responses for each eucalypt tree species under both current and future climate growth conditions are
stronger and shifted to higher leaf temperatures than the MEGAN2.1 default response (Figure 2). The peaks in current climate
$\gamma$T are 40-90 % higher than default MEGAN, whilst the peaks in future climate $\gamma$T are 45-200 % higher. The position of
the peaks are also shifted towards higher temperature optimums, by approximately 4-9 K. For the current climate growth
treatment results, running MEGAN with default settings would underestimate $\gamma$T and subsequently the isoprene emission at
leaf temperatures greater than 303 K. MEGAN assumes that at growth temperatures lower than the standard conditions, the
amplitude of the temperature response ($E_{opt}$) is lowered and the peak of that response is shifted to a lower temperature ($T_{opt}$).
These new data show for all species studied, at each growth temperature, that this is not necessarily true. Our measurements
also indicate that eucalypts have evolved to cope with the high Australian temperatures and can continue to protect against heat
damage via isoprene emission until ~320 K. Tree species with a wide geographical coverage such as *E. camaldulensis* may
also be better adapted to surviving climate change (González-Orozco et al., 2016).

Each tree in each temperature treatment group produces a similar response (numbers of trees and their temperatures at
maximum $\gamma$T given in Table 2). In the current climate-grown trees the temperature optimum in $\gamma$T is 317 - 318 K for *E.*





*tereticornis* and *E. smithii* decreasing at higher leaf temperatures. Both *E. camaldulensis* and *E. botryoides* persist at high $\gamma T$ until 328 K when measurements stopped. In the future climate-grown trees the $\gamma T$ peak is also ~317 K and there is a different response of *E. camaldulensis* and *E. botryoides* compared to the other species. $\gamma T$ in *E. camaldulensis* increases steeply with increasing leaf temperature until 321.5 K thereafter decreasing sharply. This response is common amongst the five *E. camaldulensis* in the future climate treatment, although there is scatter around this fitted response. The *E. camaldulensis*

result will dominate the weighted variables used in the modelling because of its larger geographic distribution (Table 1). We discuss the impact of this sharp downturn in $\gamma T$ at high temperatures in section 3.3.

### 3.2 Isoprene emission rates

The basal isoprene emission rates (BER) in $\mu g\ g^{-1}\ h^{-1}$ were measured at the standard 303 K and 1000 $\mu mol\ m^{-2}\ s^{-1}$ PAR (Table 2). As the current climate growth treatment represents current day climatic conditions, we only compare these with

measurements made previously on the same species. The *E. tereticornis* BER measurements are very similar to that made by Nelson et al. (2000) though lower than that reported by Jiang (2020); however our *E. camaldulensis* BER measurements are around 10 $\mu g\ g^{-1}\ h^{-1}$ higher than that listed by Benjamin et al. (1996), and our *E. botryoides* BER measurements are ~37 $\mu g\ g^{-1}\ h^{-1}$ higher than that measured by He et al. (2000). He et al. (2000) used a mixture of young and mature leaves in their experiments. Our measurements put the four eucalypt species into the high emission category.

To create new isoprene emission factor maps suitable for the modelling, we convert the BERs into landscape emission factors (LEF$_{\text{isop}}$). The average BER for each growth treatment is weighted according to their geographical areas in Table 1. BERs are then converted into LEFs using the leaf mass per unit area (LMA) in g m$^{-2}$ and scaled with LAI in m$^2$ m$^{-2}$, similar to Emmerson et al. (2018). The isoprene emission factor for trees in each temperature treatment is given by tree_EF$_{\text{isop}}$:

$$tree\_EF_{\text{isop}} = BER \times LAI \times LMA \qquad (6)$$

In the C-CTM, northern Australian vegetation is represented by broadleaf shrubs (30 – 40 %) and C4 grasses (50 to 80 % in some locations). If the isoprene emission factor maps are only based on the new eucalypt BERs, these are unlikely to be representative of shrubs and grasses. Here we ensure the non-tree fraction of grid cells in Australia are not impacted by these changes using the tree fraction (treefrac) from the ESA product.

$$LEF_{\text{isop}} = (tree\_EF_{\text{isop}} \times treefrac) + (orig\_EF_{\text{isop}} \times (1 - treefrac)) \qquad (7)$$

This leaves the fraction of original isoprene LEFs (orig_EF$_{\text{isop}}$) untouched for grass and shrub PFTs.

### 3.3 Impacts of changing C$_{\text{T1}}$, C$_{\text{T2}}$, T$_{\text{max}}$ and C$_{\text{eo}}$

Table 3 shows the results of fitting the MEGAN variables according to the current and future climate growth treatment data. These data are averages for the four tree species in the experiment, weighted according to their coverage in Table 1. The new average LEFs from our four eucalypt species are 31-48 % lower than the average LEF we use in the base run for the 3km

Sydney domain. These reductions fit within the 40 % estimated by Emmerson et al. (2019a).



The value fitted for $C_{T2}$ is very high in the future climate treatment compared with the default and current climate treatment values, due to the *E. camaldulensis* measurements in Figure 2. To assess whether $C_{T2}$ should be re-fitted we examine the impacts of changing each of these variables one at a time using a MEGAN boxmodel designed in Jiang (2020). As the impacts of the new measurements are strongest at higher temperatures, we assume conditions from the hottest day in the MUMBA campaign

(January 18th). The MEGAN boxmodel runs for 24 hours, and the results given as percentage changes to the maximum isoprene emission in Table 3. For the given fitted values on this day, the $C_{T1}$ variable has the least and $C_{eo}$ has the most impact on isoprene emissions. The high $C_{T2}$ value in the future climate treatment incurs a 19 % decrease in isoprene, which is small compared with the 282 % increase caused by $C_{eo}$. Individually $C_{eo}$ has the greatest impact on isoprene emissions but is regulated by increasing $T_{max}$ when used in tandem with other variables. However, when all variables operate together the

overall impact is an ~80 % increase in isoprene emissions for both current and future climate growth conditions. Inclusion of the average LEF reduces the maximum isoprene emission by 7 % in the current climate treatment conditions and increases by 23 % in the future climate treatment conditions on the default.

### 3.4 Model experiment set-up

Six model experiments are defined (Table 4) and are run for the periods of the field campaigns described in section 2.4. We

model the impacts of using the new current and future climate treatment temperature response variables separately from the impacts of the new LEFs on atmospheric isoprene mixing ratios. For experiments 1 to 5, we use the same hourly meteorology, current day tree distribution maps and LAI datasets to drive the C-CTM. This allows us to separate the temperature effect in isoprene emissions from other influences which may change in a future climate. The intention is to investigate changes in isoprene emissions resulting from the temperature response results, not to combine these with future land-use changes and how

the hourly meteorology will be impacted by climate change. However, in experiment 6 we use a simple delta-scaling approach to address how a future climate may impact the driving input temperatures to MEGAN.

We take the average change ($\delta$2050) in projected summertime surface temperatures for Australia under the RCP 8.5 scenario from eight models in the Coupled Model Intercomparison Project 5 (CMIP5) (for details see supplementary). We only scale the surface temperature, thus experiment 6 is not a 2050 representation of the whole atmosphere. This restricts the use of the

delta-scaled temperatures as a MEGAN input and not the temperature used for chemical reactions, as mass balance difficulties would occur by not also delta-scaling the pressure and air density through the height of the atmosphere. We estimate the reaction rate of isoprene with OH (calculated as $2.54 \times 10^{-11}$ exp(410/T) in MOZART-T1) would decrease by 1.7 % with the 3.5 K temperature rise between our current and future climate growth treatments.

Our implied future climate does not include the associated increases in $CO_2$ mixing ratios which would decrease isoprene

emissions (Heald et al., 2009). When future changes in land use and $CO_2$ are considered, Sharkey and Monson (2014) suggest that net isoprene emissions will increase due to increasing temperature. Also, our future climate temperature treatment was not conducted in a higher $CO_2$ atmosphere, so the model experiment is consistent.

Delta-scaling adds ~2 K to the surface temperatures near Sydney, resulting in an almost doubling of the number of hours above 303 K in the Bringelly and SPS1 field campaigns (74 and 53 hours, respectively), however none are pushed into the 313





K category. At MUMBA which already had six hours above 313 K, the number of hours in this category more than doubles to
13.

If the leaf temperature is varied within Equations 1-4 and $\gamma$T is multiplied by a normalised LEF, the impacts of experiments
1-5 on isoprene emission start at about 283 K (Figure 3). Experiment 6 follows the FC_$\gamma$T+LEF profile. Here, the new experi-
mental LEFs are normalised to the default MEGAN LEF. The default MEGAN profile has a peak isoprene emission at 311 K.
The CC_$\gamma$T and FC_$\gamma$T experiments cause the isoprene emission peak to shift to 324 K, with three times the default emission
value. The sharp downturn in isoprene emission in the FC_$\gamma$T and FC_$\gamma$T+LEF experiments after 324 K are due to the high $\gamma$T
of *E. camaldulensis* depicted in Figure 2. However, these results will not impact the C-CTM runs as no hourly temperature in
our three field campaigns exceeds 317 K. Most of the impacts on the C-CTM runs will occur in the 288 - 308 K range. Whilst
there is a very small decrease in the CC_$\gamma$T response compared with the default MEGAN profile at temperatures less than 300
K, overall we expect more isoprene to be emitted in the CC_$\gamma$T and FC_$\gamma$T experiments over the default MEGAN profile.
While it is intuitive from Figure 3 to expect less isoprene will be emitted in the CC_$\gamma$T+LEF and FC_$\gamma$T+LEF experiments
over the base run, this may not be the case. The LEFs used in Figure 3 are based on the domain spatial average value, however
the LEFs in experiments 3 and 5 are based on the distribution of LAI from equation 6, whilst experiments 1, 2 and 4 use the
original MEGAN LEF distribution. The current and future climate average LEFs certainly show a sustained isoprene decrease
below 314 K and 311 K respectively. Distance from source to receptor, transport and dilution will all impact results, and are
determined by running the C-CTM.

### 3.5   C-CTM results

The C-CTM is compiled with changes to MEGAN implemented according to Table 3, and run for experiments 1-6 (Table
4). The range in modelled diurnal profiles of isoprene are compared to the mean diurnal observations taken at each field
campaign (Figure 4). The CC_$\gamma$T variables only increase the isoprene mixing ratios when temperatures exceed 303 K. This has
changed the shape of the diurnal profiles of each field campaign in different ways, but generally the CC_$\gamma$T and CC_$\gamma$T+LEF
experiments have increased predicted statistical fits when compared with the base run. In MUMBA, the CC_$\gamma$T increases
the isoprene mixing ratios above the base run between 11:00 and 17:00 AEDT in the heat of the day. Very hot temperatures
during the day can often be accompanied by strong gusty winds from the Australian interior. The hottest campaign day, 18th
January 2013 during MUMBA was associated with the highest average hourly wind measurement of 8 m s$^{-1}$. Hot and windy
conditions would cause lots of sun-flecking within the tree canopy, causing very sudden temperature spikes on the leaf surface.
Physiologically, the increased production of isoprene at these times may help mediate the impacts of these sudden temperature
and light spikes, above and beyond leaf cooling via transpiration processes (Sharkey et al., 2008).

During all campaigns the CC_$\gamma$T results have decreased the isoprene from the base runs in the morning between 08:00 and
11:00 AEDT, because these temperatures are less than 303 K where the $\gamma$T are less than the default MEGAN profile (Figure
3). The CC_$\gamma$T+LEF experiments represent current day conditions, with roughly the correct magnitude of predicted isoprene
and best statistical fit compared with the observations. The FC_$\gamma$T+LEF experiment has produced more daytime isoprene than
the base run contrary to the prediction in Figure 3, because the distribution of isoprene LEFs near the field campaign sites is





different to the default MEGAN LEFs. The climate2050 experiment adds between 110 -170 % more isoprene during the day,

or approximately 2 ppb.

The MUMBA and SPS1 base diurnal profiles show too much isoprene in the model overnight compared to observed mean values, particularly in the period midnight to 06:00 AEDT. This is because there is more isoprene in the model atmosphere than was quenched by the OH radical before the OH production ceased at sundown. The isoprene becomes more concentrated at the surface because of the reduced boundary layer height; the apparent increase between midnight and 03:00 AEDT is not due

to night-time isoprene emissions. While there are few measurements of isoprene during these pre-dawn periods, it is unlikely isoprene is present. Only when daytime isoprene is reduced in the CC_$\gamma$T+LEF experiment do we see the apparent night-time isoprene is decreased.

We investigate the spatial changes to isoprene, $O_3$ and biogenic SOA in an implied future by subtracting results from the CC_$\gamma$T+LEF experiment from the climate2050 experiment during the period of the SPS1 campaign (Figure 5). These emis-

sions, mixing ratios and aerosol concentrations represent campaign averages from SPS1. We also show the smaller differences found between the FC_$\gamma$T+LEF and CC_$\gamma$T+LEF runs. The climate2050 experiment causes up to 5.2 mg m$^{-2}$ h$^{-1}$ in isoprene emissions to the immediate north of Sydney (Figure 5d), but there are also increases in the north of Australia (Figure 5c). The largest changes of 15.8 ppb in isoprene occur in sparsely inhabited northern Australia (Figure 5g), and in urbanised pockets to the south and east, where Sydney is located. The urbanisation becomes important when the increased isoprene reacts with $NO_X$

in the atmosphere causing a peak 9 ppb increase to $O_3$ near Sydney with the climate2050 differences (Figure 5l). However, the FC_$\gamma$T+LEF differences (Figure 5i) show a 0.5 ppb decrease in $O_3$ in northern Australia via quenching by the additional isoprene. Few inhabitants reside in northern Australia, meaning $O_3$ production via anthropogenic $NO_X$ is minimised. Soil $NO_X$ emissions are low in northern Australia as agricultural practices largely occur in the south east and south west of Australia. The $O_3$ deficit is still visible in the very north east of Australia in the climate2050 difference run (Figure 5k). The increase in

biogenic SOA occurs mainly in the north of Australia where up to 0.21 $\mu$g m$^{-3}$ more aerosol is predicted by the climate2050 experiment than the CC_$\gamma$T+LEF experiment (Figure 5o).

The size fraction of most secondary organic aerosol fits within the $PM_{2.5}$ classification, defined as particulate matter with an aerodynamic diameter less than 2.5 $\mu$m. Australia sets National Environmental Protection Measures (NEPMs) for $PM_{2.5}$ and $O_3$ to ensure a healthy standard of air quality for the population. The NEPM for $O_3$ is 100 ppb as a 1-hour average, and

25 $\mu$g m$^{-3}$ as a 24-hour average for $PM_{2.5}$, with a goal of reducing the $PM_{2.5}$ limit to 20 $\mu$g m$^{-3}$ by 2025. We examine the increases brought about by climate induced isoprene in the two cities impacted by most of these changes, Sydney and Darwin, in Australia's north (Figure 6).

The air quality index (AQI = NEPM/pollutant concentration x 100) in Sydney and Darwin is classed as 'very good' (AQI <33) for both pollutants, with an improving trend for $O_3$ but a declining trend for $PM_{2.5}$ (Keywood et al., 2016). Darwin is

a small city, and the biogenic component of $O_3$ changes are less than 2 ppb. However peak $O_3$ in Sydney increases by 10 – 15 ppb as an hourly average in the FC_$\gamma$T+LEF differences, but by as much as 15 - 21 ppb in the climate2050 differences (Figure 6a,b). These increases represent 10 - 21 % of the $O_3$ NEPM, and show that by doing nothing (e.g. tree type and coverage or air quality policies do not change) and allowing the temperatures to rise, large cities will likely encounter more





NEPM exceedances. The solution is not to remove native trees as they provide social amenity and have cultural significance for indigenous populations. Rather, their emissions must be accommodated via atmospheric $NO_X$ reductions. However new urban developments should consider the BVOC emission potential of prospective trees before planting (Paton-Walsh et al., 2019).

The SOA from isoprene is a small fraction of the $PM_{2.5}$ limit (shown here as 24-hour averages), though of the BVOC aerosol yields, isoprene is not expected to dominate. The aerosol yields from monoterpenes are 10-20 times higher than the isoprene yield and the monoterpene emission would increase in a warming climate (not investigated here). The climate2050 differences show days with an increase of 0.41 $\mu$g m$^{-3}$ in Sydney and 0.13 $\mu$g m$^{-3}$in Darwin (2 % and 1 % of the $PM_{2.5}$ 2025 NEPM, respectively).

## 4 Conclusions

We have measured the isoprene emission response to controlled increases in temperature from four eucalypt species, two of which have a large geographical growing extent in Australia. The trees were grown in temperatures representing the current climate summertime conditions in Australia and in temperatures representing the projected summertime conditions of +3.5 K warming under the business as usual RCP 8.5 scenario. Climate projections for Australia forecast increases in average temperatures with an accompanying increase in the frequency of extreme heatwave days (Bureau of Meteorology and CSIRO, 2018). This will likely increase in the number of days above 303 K.

The current condition experiments demonstrated a change in the isoprene emission response to temperature as compared with the default parameterisation in MEGAN. This is not a surprise, as MEGAN is built to represent a range in ecosystem responses, but may go some way to explain why difficulties have been encountered when modelling isoprene in Australia previously. Both the current and future climate growth treatment temperature responses shifted the peak in $\gamma$T by 4-9 K, signifying that these four eucalypt species were observed to continue emitting isoprene until well past the default maximum temperature for emission at 313 K. This suggests the eucalypts used in this study have evolved to protect against higher temperatures as expected with climate change.

Higher basal emission rates were measured in three of the eucalypt species in our experiment than have been previously measured. However, the conversion of these average weighted emission rates to LEFs for use in the C-CTM, resulted in a lower average LEF than are currently being used in the base run. This is due to low biomass measured on our leaves, and because the isoprene emission factors from regions described as shrubs or grasslands were not altered. The spatial distribution of the new LEFs were based on the LAI distribution, different to the default MEGAN isoprene LEF map.

The model results using the new current climate growth temperature responses improved the statistical fits of the diurnal profiles compared to the measurements in average isoprene across our three field campaign periods. The overall magnitude of the modelled profile was also brought into better agreement with observations in combination with the new current climate growth LEFs. MEGANv2.1 essentially works using a series of variables dependant on vegetation type and biogenic compound emission traits, and the results here suggest that the four MEGAN variables altered in our experiments could also become ecosystem or location specific.





Despite our measurements being conducted on sapling trees which may exhibit higher isoprene emissions than adult trees, we expect the trend between the current and future climate growth emissions to be similar amongst trees of all ages. Our model experiments simulating isoprene emissions in a 2050 climate examined the differences between these runs and the

CC_$\gamma$T+LEF experiment. Two future experiments were conducted, the first using current day meteorology, and the second using a delta-scaled surface temperature change to projected 2050 summertime temperatures. The FC_$\gamma$T+LEF experiment showed increases in isoprene emissions in the north of Australia, as well as closer to Sydney. These increases led to $O_3$ rising $10 - 15$ ppb close to Sydney as a result of the increased isoprene, whilst decreasing in sparsely populated northern Australia through quenching by the additional isoprene. The climate2050 experiment showed much larger increases in isoprene, $O_3$

and biogenic SOA across Australia. Delta-scaling the surface temperatures was the simplest way of conducting future climate experiments. Future work should investigate getting a downscaled version of the 2050 atmosphere from CCAM which would provide the hourly meteorology throughout the atmosphere that the C-CTM requires.

The future is expected to bring increased temperatures, $CO_2$ and land use changes. Sharkey and Monson (2014) evaluated the isoprene trade-off in each of these scenarios and concluded the temperature effects would dominate. $O_3$ is a secondary

product of isoprene oxidation, and is currently maintained at healthy levels in Australia. In order to maintain these levels, air quality policy should investigate methods to reduce anthropogenic $NO_X$ emissions in city regions to accommodate these climate change induced increases in BVOC emissions. In addition, tree planting efforts in new urban developments should also consider the BVOC emission potential of prospective trees.

*Data availability.* The LAI data product was retrieved from MCD15A2 version 4 from the online Data Pool, courtesy of the NASA Land Processes Distributed Active Archive Center (LP DAAC), USGS/Earth Resources Observation and Science (EROS) Center, Sioux Falls, South Dakota, https://lpdaac.usgs.gov/data_access/data_pool

*Author contributions.* KME and MP devised the modelling study and wrote the manuscript. KME conducted the modelling. MP, MJA, SP and MGT conducted the experimental work. MJA, SP and MGT edited the manuscript.



*Acknowledgements.* KME thanks Christine Wiedinmyer at the University of Colorado, Boulder for assistance with the ESA landcover
product data and John Clarke at CSIRO for helpful discussions on using the climate projections data.



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





**Table 1.** Geographic range size of each *Eucalyptus* species in Australia and the isoprene emission rate by dry leaf weight basis

| Tree | Common name | Area (km$^h$)[a] | % weight | Emission category | Average emission $\mu g \ g^{-1} \ hr^{-1}$ |
|---|---|---|---|---|---|
| *E. camaldulensis* | River red gum | 6 040 600 | 86.32 | high | 16.6[b] 32.5 [c] |
| *E. tereticornis* | Forest red gum | 792 575 | 11.32 | high | 32.7[d]  38.2[e] |
| *E. smithii* | Blackbutt peppermint | 95 750 | 1.37 | unknown | - |
| *E. botryoides* | Bangalay | 74 175 | 1.06 | moderate | 5.3[b] |

[a] Species area in 2014 (from González-Orozco et al. (2016)). [b] He et al. (2000). [c] Benjamin et al. (1996). [d] Nelson et al. (2000). [e] Jiang (2020), from trees growing in ambient $CO_2$ concentrations.





**Table 2.** Average isoprene basal emission rates (BER), leaf mass per unit area (LMA) and temperature at maximum $\gamma$T from each pool of trees under the current and future climate growth conditions. Values in brackets are standard deviations. Data in right-hand column is derived from model fits.

| Treatment | Species | No. of trees | BER, $\mu$ g g$^{-1}$ hr$^{-1}$ | LMA, g m$^{-2}$ | Temp at max $\gamma$T, K |
|---|---|---|---|---|---|
| current climate | *E. tereticornis* | 6 | 29.14 (13.91) | 61.53 (5.42) | 317.8 |
| | *E. smithii* | 6 | 41.21 (17.31) | 54.93 (13.71) | 317.8 |
| | *E. botryoides* | 6 | 42.46 (23.64) | 72.51 (15.25) | 318.4 |
| | *E. camaldulensis* | 6 | 42.87 (22.87) | 72.79 (6.14) | 322.1 |
| future climate | *E. tereticornis* | 6 | 41.57 (28.08) | 64.05 (9.58) | 317.3 |
| | *E. botryoides* | 5 | 55.18 (27.27) | 77.96 (12.55) | 317.5 |
| | *E. smithii* | 6 | 61.61 (20.01) | 58.08 (5.10) | 317.0 |
| | *E. camaldulensis* | 5 | 66.95 (22.44) | 73.18 (4.64) | 321.5 |





**Table 3.** Changes to MEGAN variables based on fitted data from current and future climate growth experiments. Percentages in brackets indicate change in maximum daily isoprene emissions due to change in variable. *Value of average LEF from the inner 3 km domain.

| | MEGAN2.1 | current climate growth treatment | future climate growth treatment |
|---|---|---|---|
| Average LEF ($\mu g\ g^{-1}\ hr^{-1}$) | 9491 * | 4919 (-48%) | 6585 (-31 %) |
| $C_{T1}$ | 95 | 110.55 (-1 %) | 75.04 (+1 %) |
| $C_{T2}$ | 230 | 167.11 (+5 %) | 1158.36 (-19 %) |
| $T_{max}(K)$ | 313 | 325 (-55 %) | 323 (-46 %) |
| $C_{eo}$ | 2 | 6.77 (+238 %) | 7.69 (+282 %) |
| All variables without LEF | | +81 % | +76 % |
| All variables + LEF | | -7 % | +23 % |





**Table 4.** Description of each model experiment. CC = current climate, FC = future climate.

| Experiment | Name | Emission factors | Temperature response | Meteorology used to drive MEGAN |
|---|---|---|---|---|
| 1 | Base | default | default | current |
| 2 | CC_$\gamma$T | default | fitted CC | current |
| 3 | CC_$\gamma$T+LEF | CC LEF | fitted CC | current |
| 4 | FC_$\gamma$T | default | fitted FC | current |
| 5 | FC_$\gamma$T+LEF | FC LEF | fitted FC | current |
| 6 | Climate2050 | FC LEF | fitted FC | current + $\delta$2050 |

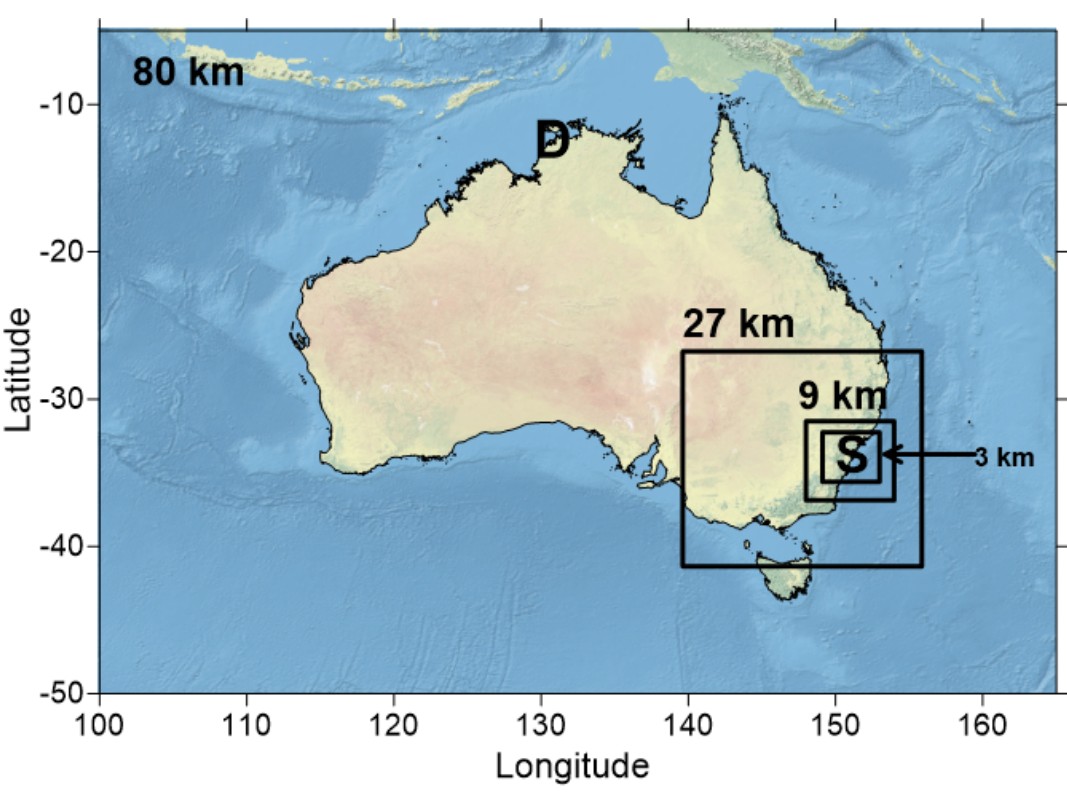

**Figure 1.** Map to show nests of model domains from 80 km Australia-wide to 3 km inner Sydney domain. 'S' and 'D' mark the locations of Sydney and Darwin, respectively.



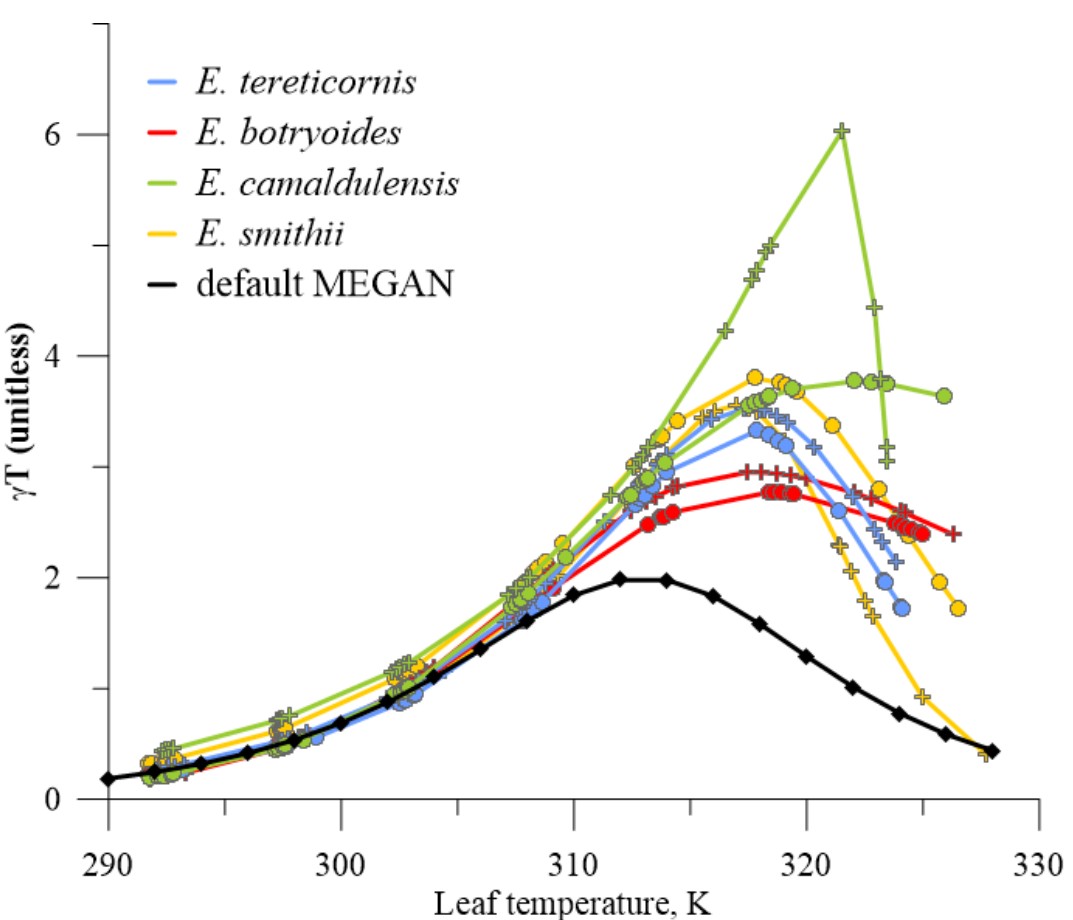

**Figure 2.** Comparison of $\gamma$T with leaf temperature calculated using default values in MEGAN to results from four eucalypt tree species under current climate (filled circles) and future climate (+ sign) growth conditions.

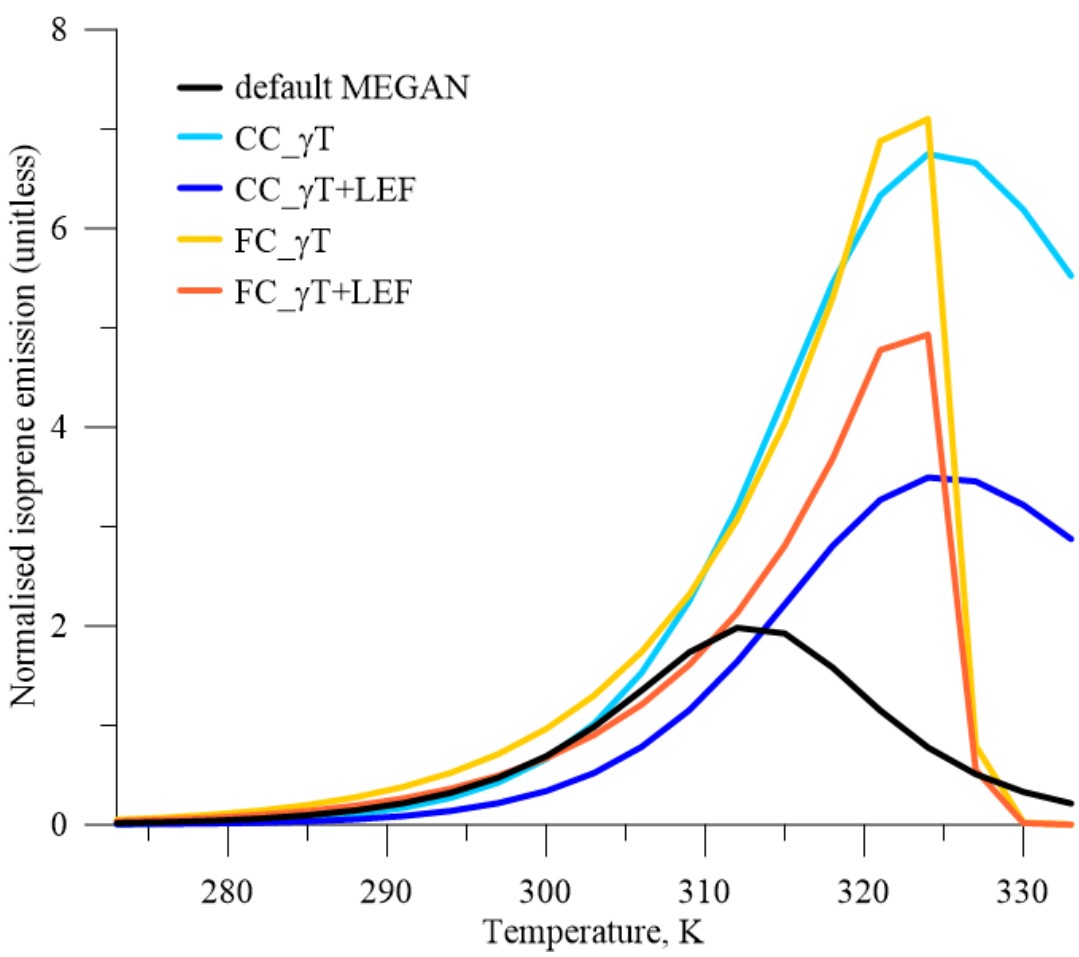

**Figure 3.** Impacts of new MEGAN variables on normalised isoprene emission rates at increasing ambient temperatures.





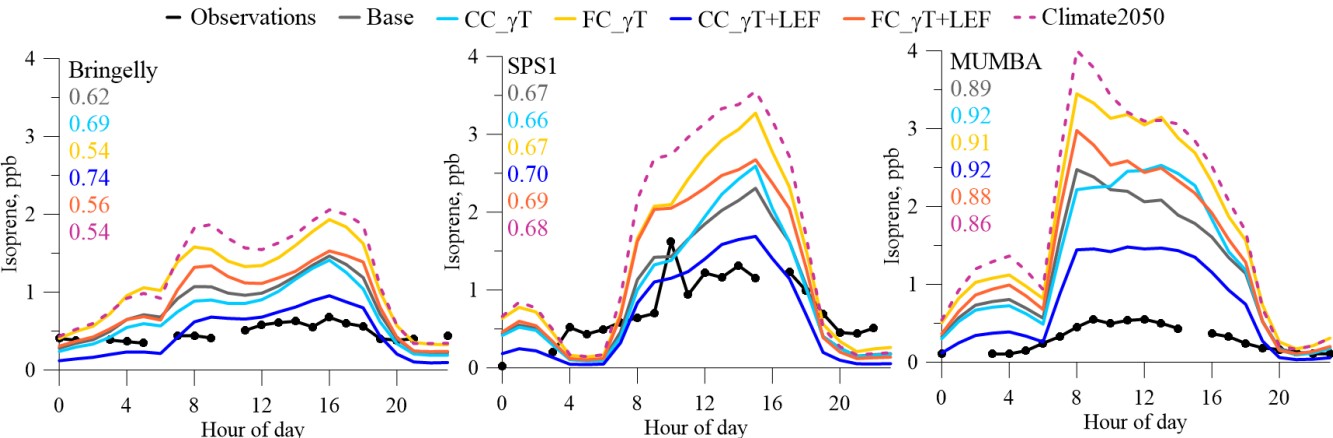

**Figure 4.** Average diurnal time series in isoprene mixing ratios incurred by the different model experiments at each field campaign site. $r^2$ values between modelled and observed isoprene given in same colours as legend.



**Figure 5.** Difference between FC_$\gamma$T+LEF and CC_$\gamma$T+LEF runs (panels a, b, e, f, i, j, m and n) during the SPS1 campaign. The difference between the climate2050 runs and CC_$\gamma$T+LEF runs are shown in panels c, d, g, h, k, l, o and p. Left to right, panels a-d: Isoprene emission, panels e-h: isoprene mixing ratio, panels i-l: ozone mixing ratio and panels m-p: biogenic SOA concentration in Australia at 80 km and Sydney at 3 km domains.



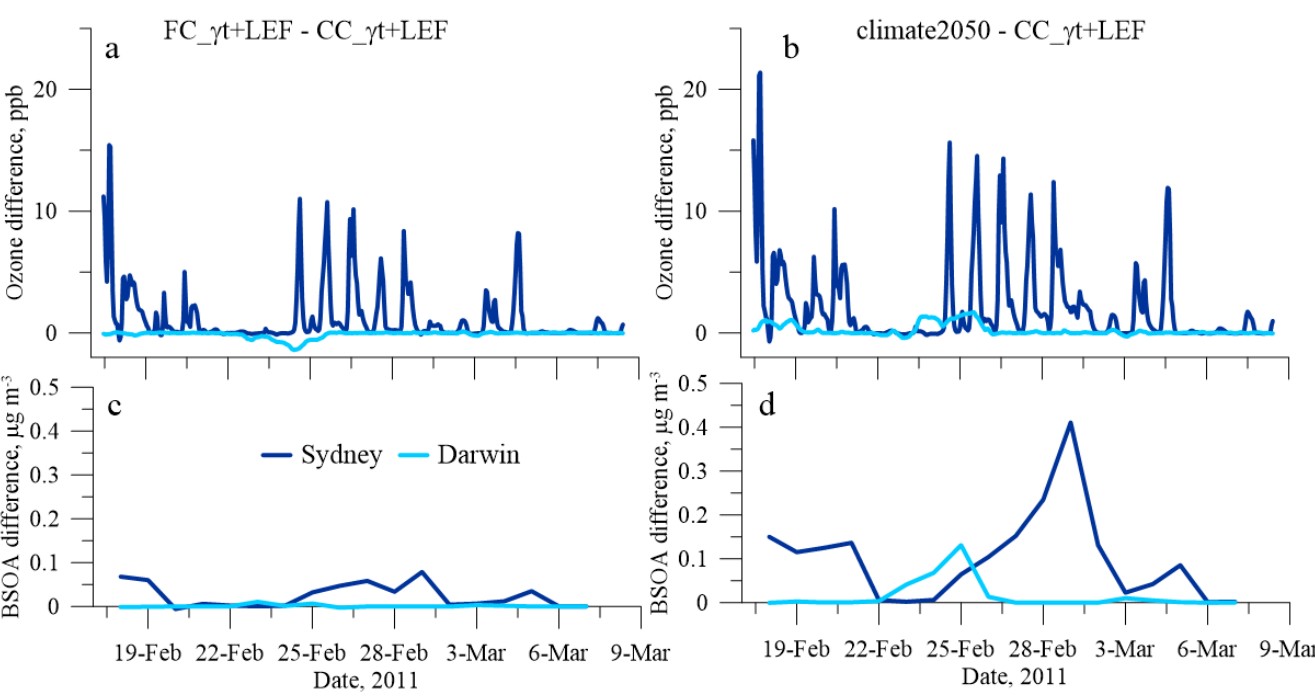

**Figure 6.** Differences in hourly ozone (panels a and b) and biogenic secondary organic aerosol (panels c and d) at Sydney and Darwin during SPS1 duration. Left panels (a and c) show FC_$\gamma$T+LEF - CC_$\gamma$TLEF whilst right panels (b and d) show climate2050 - CC_$\gamma$T+LEF.