# Peer review of "Temperature response measurements from eucalypts give insight into the impact of Australian isoprene emissions on air quality in 2050"

_Atmospheric Chemistry and Physics, 2020_

## Referee Comment (RC1) · Anonymous Referee #1 · 10 Mar 2020

I have enjoyed reading the manuscript and find it a remarkable piece of work, stretching from individual measurements to continental scale impact estimates. At each level, interesting results are presented, starting with uncertainties connected with assumptions regarding emission potential and temperature sensitivity in standard emission models. The analysis is timely and thorough although the literature overview could be a bit more comprehensive. The text is well written and the conclusions are sound. My only concern is that disregarding the CO2 effect in the scenario analyses might call for exaggerated calls for action. Thus, I would appreciate a 7th simulation to account for this, e.g. a scenario such as the 6th run but with increased CO2, despite the fact that I am aware of the inconsistency with the measurements (and also with some uncertainties

related to a shift of temperature response curves under higher CO2 as shown by Sun et al.).

From the few specific remarks, I would like to stress the benefits from an improved literature overview (i.e. L25ff). The generally high emission potential of eucalypts in comparison with other species have firstly been depicted in Evans et al. 1982 and can also been derived from Kesselmeier and Staudt – although the values concentrate on E.globulus. Karlik and Winer as well as Geron et al. provide an additional emission rate of E.camaludensis (28, 14.6, add to Table 1) and a couple of other eucalypt species – although not under Australian conditions.

You may also note that different temperature responses and emission factor variability have been obtained before, starting with the original Guenther et al. 1991 publications and widely discussed e.g. in Niinemets et al. 2010 (e.g. L44ff).

In the end of chapter 2.2 (93ff), I got the impression that the authors are carried away a bit. First, the last paragraph seems to fit better into a discussion; and second, the first sentence is not logical (the measurements are hardly going to change but emission rates and species abundance probably will). By the way, I am still uncertain to which degree these 4 eucalypt species are actually representative for the Australian forests or how abundant the are in relative terms (L163).

With one sole reference, the protective functions of isoprene to sun flecks and very high temperatures are not very well acknowledged (L268ff). There are several publications (e.g. Behnke et al.) and reviews (Loreto and Fineshi) that illustrate this function. In fact, emission is prolonged even under carbon deficit conditions (Yanez-Serrano et al.).

Finally, the thought came to my mind that instead of removing the trees (which is of course not recommended), the forest management might be compelled to introduce species others than Eucalypts that are not emitting isoprene (L314ff). However, given the protective function mentioned above, this option might not be advisable because non-emitters might not be able to withstand the coming heat (Penuelas and Munne

Bosch, Ryan et al.).

Indicated references

Behnke, K., Loivamäki, M., Zimmer, I., Rennenberg, H., Schnitzler, J.P. and Louis, S. 2010 Isoprene emission protects photosynthesis in sunfleck exposed Grey poplar. Photosynth. Res., 104 (1), 5-17.

Evans, R.C., Tingey, D.T. and Gumpertz, M.L. 1982 Estimates of isoprene and monoterpene emission rates in plants. Botanical Gazette, 143, 304-310.

Geron, C., Harley, P. and Guenther, A. 2001 Isoprene emission capacity for US tree species. Atmos. Environ., 35 (19), 3341-3352.

Guenther, A.B., Monson, R.K. and Fall, R. 1991 Isoprene and monoterpene emission rate variability: Observations with Eucalyptus and emission rate algorithm development. J. Geophys. Res., 96 (D6), 10799-10808.

Karlik, J.F. and Winer, M. 2001 Measured isoprene emission rates of plants in California landscapes: comparison to estimates from taxonomic relationships. Atmos. Environ., 35 (6), 1123-1131.

Kesselmeier, J. and Staudt, M. 1999 Biogenic volatile organic compounds (VOC): An overview on emission, physiology and ecology. J. Atmos. Chem., 33 (1), 23-88.

Loreto, F. and Fineschi, S. 2015 Reconciling functions and evolution of isoprene emission in higher plants. New Phytol., 206 (2), 578-582.

Niinemets, Ü., Arneth, A., Kuhn, U., Monson, R.K., Peñuelas, J. and Staudt, M. 2010 The emission factor of volatile isoprenoids: stress, acclimation, and developmental responses. Biogeosciences, 7 (7), 2203-2223

Peñuelas, J. and Munne-Bosch, S. 2005 Isoprenoids: an evolutionary pool for photoprotection. Trends Plant Sci., 10 (4), 166-169.

Ryan, A.C., Hewitt, C.N., Possell, M., Vickers, C.E., Purnell, A., Mullineaux, P.M. et al. 2013 Isoprene emission protects photosynthesis but reduces plant productivity during drought in transgenic tobacco (Nicotiana tabacum) plants. New Phytol., 201 (1), 205–216.

Sun, Z., Hüve, K., Vislap, V. and Niinemets, Ü. 2013 Elevated [CO2] magnifies isoprene emissions under heat, alters environmental responses and improves thermal resistance in hybrid aspen. J. Exp. Bot., 64 (18), 5509-5523

Yáñez-Serrano, A.M., Mahlau, L., Fasbender, L., Byron, J., Williams, J., Kreuzwieser, J. et al. 2019 Heat stress causes enhanced use of cytosolic pyruvate for isoprene biosynthesis. J. Exp. Bot., 70 (20), 5827-5838.
* * *

---

## Referee Comment (RC2) · Anonymous Referee #2 · 30 Mar 2020

[referee-annotated manuscript omitted]

---

## Author Comment (AC1) · 6 Apr 2020

**Response to reviewer #1**

We thank reviewer #1 for the positive assessment of our work. Reviewer comments are in italics.

*I have enjoyed reading the manuscript and find it a remarkable piece of work, stretching from individual measurements to continental scale impact estimates. At each level, interesting results are presented, starting with uncertainties connected with assumptions regarding emission potential and temperature sensitivity in standard emission models. The analysis is timely and thorough although the literature overview could be a bit more comprehensive. The text is well written and the conclusions are sound. My only concern is that disregarding the CO2 effect in the scenario analyses might call for exaggerated calls for action. Thus, I would appreciate a 7th simulation to account for this, e.g. a scenario such as the 6th run but with increased CO2, despite the fact that I am aware of the inconsistency with the measurements (and also with some uncertainties C1 related to a shift of temperature response curves under higher CO2 as shown by Sun et al.).*

The $CO_2$ effect is easy to add, and we have included a 7$^{th}$ simulation as suggested.

The text at line 239 has been changed to say: "The climate2050 run does not include the associated increases in $CO_2$ mixing ratios, to be consistent with our measurements which were also not conducted in a higher $CO_2$ atmosphere. A 7th simulation assumes a 550 ppm $CO_2$ atmosphere on top of the delta-scaled surface temperatures, employing Heald et al's (2009) method for calculating short and long term $CO_2$ activity factors, $\gamma C$. Fixing the atmospheric $CO_2$ to 550 ppm reduces the isoprene emissions by 5% in the short term and 13% in the long term."

An extra column has been added to table 4 to show simulation 7 has $\gamma C$ added.

Add results from this experiment to line 280 "The addition of a higher $CO_2$ atmosphere has reduced the daytime isoprene by 15 - 26 % from the climate2050 run, across the three campaigns."

To avoid too many panels in figure 5, we'll leave the climate 2050 run as the upper end to the range in results. However, figure 6 now contains the time series from the climate2050_$\gamma C$ run. Figure 6 has been altered to have the Sydney results on the left and the Darwin results on the right.

Change text at line 310 "However peak $O_3$ in Sydney increases by 10 – 15 ppb as an hourly average in the FC_$\gamma T$+LEF differences, but by 12 - 17 ppb in the climate2050_$\gamma C$ differences and 15 - 21 ppb in the climate2050 differences (Figure 6a,b). These increases represent 10 - 21 % of the $O_3$ NEPM."

Change the text at line 320 "The climate2050 (and $\gamma C$) differences show days with an increase of 0.42 $\mu g\ m^{-3}$ in Sydney and 0.14 $\mu g\ m^{-3}$ in Darwin (2 % and 1 % of the $PM_{2.5}$ 2025 NEPM, respectively)."

Make minor changes to conclusions at line 350 "Three future experiments were conducted, the first using current day meteorology, the second using a delta-scaled surface temperature change to projected 2050 summertime temperatures, and the third using a 550 ppm atmospheric $CO_2$ on top of the delta scaled temperatures."

And at line 354. "The climate2050 experiment showed much larger increases in isoprene, $O_3$ and biogenic SOA across Australia, tempered slightly by the addition of increased atmospheric $CO_2$."

Also adjust abstract at line 13 to include "A 550 ppm $CO_2$ atmosphere in 2050 mitigates these peak Sydney $O_3$ mixing ratios by 4 ppb. Nevertheless, these forecasted increases in $O_3$ are up to one fifth..."

*From the few specific remarks, I would like to stress the benefits from an improved literature overview (i.e. L25ff). The generally high emission potential of eucalypts in comparison with other species have firstly been depicted in Evans et al. 1982 and can also been derived from Kesselmeier and Staudt – although the values concentrate on E.globulus. Karlik and Winer as well as Geron et al. provide an additional emission rate of E.camaludensis (28, 14.6, add to Table 1) and a couple of other eucalypt species – although not under Australian conditions.*

Add text at line 26 "Native to Australia, eucalypt trees are amongst the highest BVOC emitters of any plant species (Benjamin et al., 1996; Evans et al., 1982; Kesselmeier and Staudt, 1999)"

Add *E. camaldulensis* emission rates from Karlik and Winer (2001) to table 1. The value listed in the Geron et al (2001) reference is the measurement by He et al (2000) already listed in table 1. Geron et al convert He et al's measurement to a dry weight of carbon (ug C $g^{-1}$ $h^{-1}$) which is why the value is slightly different. Here we're using µg isoprene $g^{-1}$ $h^{-1}$.

*You may also note that different temperature responses and emission factor variability have been obtained before, starting with the original Guenther et al. 1991 publications and widely discussed e.g. in Niinemets et al. 2010 (e.g. L44ff).*

True, and I noted from Guenther et al (1993) when one of the early models for isoprene emission with temperature was defined, it was based on "empirical coefficients which were determined by nonlinear best fit procedures using eucalyptus, sweet gum, aspen, and velvet bean emission rate measurements."
Change text at line 45 "Whilst the MEGAN parameterisations are fitted from a wide range of ecosystem responses to environmental conditions, there are spatial and temporal exceptions to these standards which are comprehensively reviewed by Niinemets et al (2010)"

*In the end of chapter 2.2 (93ff), I got the impression that the authors are carried away a bit. First, the last paragraph seems to fit better into a discussion; and second, the first sentence is not logical (the measurements are hardly going to change but emission rates and species abundance probably will). By the way, I am still uncertain to which degree these 4 eucalypt species are actually representative for the Australian forests or how abundant the are in relative terms (L163).*

We've decided to drop the first part of this paragraph and include more details of how the eucalypt species are spread in the earlier part of section 2.2. We also include species occurrence maps from the Atlas of Living Australia in the supplementary section, reproduced below.

At line 75 "*E. camaldulensis* and *E. tereticornis* have a wide geographical representation within Australia, with a latitudinal native growing range of 9-38 °S (Atlas of Living Australia, 2019), (supplementary figure S1). *E camaldulensis* is the most widely naturally distributed species of all eucalypts in Australia (Atlas of Living Australia, 2019). The native climatic distribution range of *E. botryoides* and *E. smithii* are restricted to the south east coastal regions. All four species are forecast to exist in future, but only *E. camaldulensis* is predicted to expand its growing area by 2085 (González-Orozco et al., 2016). "

[Figure]

**Figure S1 clockwise from top left.** *E camaldulensis*, *E tereticornis*, *E smithii*, *E. botryoides*.

*With one sole reference, the protective functions of isoprene to sun flecks and very high temperatures are not very well acknowledged (L268ff). There are several publications (e.g. Behnke et al.) and reviews (Loreto and Fineshi) that illustrate this function. In fact, emission is prolonged even under carbon deficit conditions (Yanez-Serrano et al.).*

At line 270 "Hot and windy conditions would cause lots of sun-flecking within the tree canopy, causing sudden temperature spikes on the leaf surface. Physiologically, the increased production of isoprene during temperature and light spikes helps to maintain photosynthesis   during times of mild stresses (Loreto and Fineschi, 2015), above and beyond leaf cooling via transpiration processes (Sharkey et al., 2008). High isoprene emitters can better survive prolonged heatwaves (Yáñez-Serrano et al., 2019), although Aspinwall et al's (2019) study on our four eucalypt species showed trees grown under future climate conditions suffered greater heatwave damage than the same species in current climate conditions."

*Finally, the thought came to my mind that instead of removing the trees (which is of course not recommended), the forest management might be compelled to introduce species others than Eucalypts that are not emitting isoprene (L314ff). However, given the protective function mentioned above, this option might not be advisable because non-emitters might not be able to withstand the coming heat (Penuelas and Munne-Bosch, Ryan et al.).*

We're not recommending eucalypt trees are removed, though will add a sentence about non emitters being unable to cope with heat stresses.

[revised manuscript text omitted]

---

## Author Comment (AC2) · 6 Apr 2020

**Response to reviewer #2**

We thank Reviewer #2, who added their comments in the form of an annotated PDF document. We hope we caught them all below! Reviewer comments are in italics.

*Discussion on use of saplings in the discussion. I think this needs to come back even more in the discussion - what impact then would this have on your findings/conclusions?*
We have added discussion at line 347 (refer to penultimate comment)

*Picture of the leaf cuvette with further details on experiment. A picture of this might help as I am not familiar with it. Does the cuvette cover the full leaf? Also, how long was the cuvette exposed? Was this done in-line with PTR-MS? So continuously? Or every 7-minutes?*
We feel a picture of a commercial leaf cuvette doesn't add to the manuscript. However, we have added more details to explain that gas exchange measurements were continuous and that the full leaf was measured. Leaf area measurements are already described in L108-9 of the original submission.

At line 98: "Leaf gas exchange measurements were made continuously with a LI-6400XT portable photosynthesis system (Li-Cor Inc., Lincoln, NE,USA) connected to a Walz 3010-GWK1 leaf cuvette (maximum surface area for leaf 140 cm$^2$; Heinz Walz GmbH, Effeltrich, Germany)."

*Were the results of these 5-6 trees quite similar? So we can assume this number of points is representative of each species?*
The values obtained from the 5-6 replicated were similar even though there was intrinsic variability within a species. To demonstrate this we have added a figure to the supplementary section that highlights the trends in response quite clearly.

[Figure]

Figure: Temperature response of normalised isoprene emission rate from four *Eucalyptus* species (a) *E. camaldulensis*, (b) *E. botryoides*, (c) *E. smithii* and (d) *E. tereticornis* grown under two different temperature regimes. Open circles (dotted lines) are current climate and filled circles (dashed line) are future climate. The solid line in each panel is the normalised isoprene emission calculated using default MEGAN values. Data are normalised to the isoprene emission rate measured at a leaf temperature of 303 K. Error bars (horizontal and vertical) are means ± one standard deviation of 4-6 replicate plants.

*It is not clear at this point in the text of the importance of 303K and 313K. I would recommend stating the importance of these thresholds here.*
These values relate to perceived thresholds at 30C and 40C. The point of this sentence is to convey how hot these three summers were, so will change to maximum and averages over the campaigns. Alter text at line 131 "Maximum (and average) measured temperatures were 308.9 K (295.9 K) for Bringelly, 310.0 K (295.6 K) for SPS1 and 317.2K (295.3 K) for MUMBA. Climate projections for Australia forecast increases in average temperatures with an accompanying increase in the frequency of extreme heatwave days (Bureau of Meteorology and CSIRO, 2018)."

Also remove text at line 243. Move "Delta-scaling adds ~2 K to the surface temperatures near Sydney" to line 233.

*If your emissions are lower than He who used a mixture of leaf ages, could your results be a partial artefact of not having old leaves?*
Our basal emission rates were higher than He's and are consistent with the observations of Street (1997) that younger leaves are higher emitters.

Re-write from line 193: "He et al. (2000) used a mixture of young and mature leaves in their experiments, which could be one explanation for the difference in emission rates as young leaves are expected to be higher emitters than older leaves in *Eucalyptus* (Street et al. 1997). However, as the growth conditions (particularly light and temperature) and measurement protocols between this study and He et al. (2000) were different (we directly measured BER with a leaf cuvette at 1000 $\mu$mol m$^{-2}$ s$^{-1}$ and 30 °C while He et al. used a dynamic chamber and scaled emissions to 1000 $\mu$mol m$^{-2}$ s$^{-1}$ PAR and 30 °C using algorithms from Guenther et al. (1993)), it is difficult to undertake a direct comparison. However, our measurements put the four eucalypt species into the high emission category."

*Are these averages for 4 species in MEGAN or in your data or both? I assume it can't be both as there wasn't info on all your species before. So then Megan is for the three species?*
The values given for MEGAN in table 3 is the average across the isoprene emission factor map for the inner 3km domain. This is stated in the table caption. On line 31 we state that the eucalypt tree emissions in MEGAN were based on six studies with numerous species involved. The default isoprene emission factor map represents all tree, shrub and grass species in Australia (line 201). What we have done is then alter the tree portion only of this map using our 4 tree species

measurements but weighted them according to the area each species takes up in Australia (line 208).

To make this clearer we alter text at line 207 "Table 3 shows how the results of fitting CT1, CT2, Tmax and Ceo compared to the default MEGAN values. These new fitted data are for the four tree species in the experiment, weighted according to their coverage in Table 1. The new average LEFs from our four eucalypt species are 31-48 % lower than the default average MEGAN LEF we use in the base run for the 3km Sydney domain."

*What does the 40% emission reduction relate to? estimated here for what? I would recommend giving more context to this statement.*
At line 210: "Previous modelling showed that a 40% reduction in isoprene was needed to better match the observations from our three field campaigns (Emmerson et al., 2019)."

*New numbers to default MEGAN? What are the two levels used for each variable? What do you change them to in this senstivity? Is it MEGAN vs new numbers?*
The two levels in table 3 refer to the measurements made on the current climate and future climate grown trees as described in section 2.2. The sensitivity involves changing the default MEGAN values to the two values in columns 3 and 4 of table 3.
Alter text at line 211 "The value fitted for $CT_2$ is very high (1158.36 kJ mol$^{-1}$) in the future climate treatment compared with the current climate treatment (167.11 kJ mol$^{-1}$) and default MEGAN (230 kJ mol$^{-1}$), due to the future climate *E. camaldulensis* measurements in Figure 2."

*Thus, CT2 won't be re-fitted, correct?*
Correct.
At line 217 "The high $CT_2$ value in the future climate treatment will not be refitted, as the incurred 19 % decrease in isoprene is small compared with the 282 % increase caused by $C_{eo}$."

*(relates to figure 3) what are the LEFs normalised to? Assume that these are max emission points in summer? what are the emissions normalised to? Are these the mean points? Is there deviations as this is across multiple field campaigns? As these are all Jan-Mar, I assume that is max emission period, correct?*
The LEFS do not change regardless of season. Instead they are moderated in MEGAN by the environmental factors mentioned on line 44 which accounts for seasonal differences (temperature, PAR, leaf area index, leaf age, soil moisture, and suppression via ambient $CO_2$ concentrations). The LEFS in this figure are normalised by the default MEGAN LEF from table 3. I.e., default MEGAN LEF = 1, CC_$\gamma$T+LEF is 48% less and FC_$\gamma$T+LEF is 31% less than the default.
Change the description of the figure at line 247 "If the leaf temperature is varied within Equations 1-4 and $\gamma$T is multiplied by the LEF, the impacts of experiments 1-5 on isoprene emission start at about 283 K (Figure 3). Experiment 6 follows the FC_$\gamma$T+LEF profile. Here, the new current and future climate LEFs are normalised by the default MEGAN LEF."

*due to spatial heterogeneity and proximity to study site, right? The next sentence is a little confusing of an explanation to me, so if possible I would recommend summarizing it here before explanation*

At line 256: "While it is intuitive to expect less isoprene will be emitted in the CC_γT+LEF and FC_γT+LEF experiments over the base run (from Figure 3), this may not be the case due to spatial heterogeneity in the new current and future climate LEF maps."

*Where are these decreases on Fig 3? I don't see them.*
Change text at line 259: "The results from experiments 3 and 5 certainly show a sustained isoprene decrease below 314 K and 311 K respectively."

*The colour between Obs and Base hard to see in legend. Is it possible to show also circle with line in "observation".*
Done

*Are these field campaigns at one site for each campaign? So are modelled output extracted for that one point? Or full domain?*
At line 263: "The C-CTM is compiled with changes to MEGAN implemented according to Table 3, run for experiments 1-6 (Table 4) and the isoprene time series is extracted at each field campaign site. The modelled mean diurnal profiles of isoprene are then compared to the mean diurnal observations taken at each field campaign (Figure 4)."

*As seen in fig 3?*
Correct. At line 265 "The CC_γT variables only increase the isoprene mixing ratios when temperatures exceed 303 K (from figure 3)"

*How were r2 calculated? Is this based on all hourly values for each field campaign? Were there similar number of points? Also, why are there missing points in the observations? Did you use a data completeness threshold or something to develop these?*
The gaps in the observations are at specific times for blanks/calibrations, eg 2 and 3am local time. The $r^2$ were calculated by comparing the mean diurnal modelled isoprene to the mean diurnal observed isoprene. However the $r^2$ is similar if all hourly data is included.

At line 265 "Instrument calibrations/blanks are taken at least twice a day, incurring frequent regular gaps in observed isoprene."

At line 266 "but generally the CC_γT and CC_γT+LEF experiments have increased the diurnal modelled to observed $r^2$ when compared with the $r^2$ between the base run and observations."

*Comment refers to continuing large bias in MUMBA results.*
At line 276 "The CC_γT+LEF experiments represent current day conditions, with roughly the correct magnitude (MUMBA excepted) of predicted isoprene and best statistical fit compared with the observations."

*Refers to nighttime decrease in isoprene. It is still higher than measurements in MUMBA - and then is too small 4-8:00 in SPS1. I would recommend adding some more discussion on the biases in this section. In addition, it might be helpful to show in supp materials only the best performing one(s)*

*(cc_T+LEF) with base and observations with std deviations - to show also the spread of the data and if these are within each other by 1 std dev or not.*

The biases in pre-dawn SPS1 are because there is a very slight rise in the boundary layer, causing dilution of the atmospheric isoprene:

[Figure]

At line 286 "Conversely a slight rise in the model boundary layer at 04:00 AEDT in SPS1 causes dilution of the atmospheric isoprene."

We have always been aware of high modelled biases in Australian isoprene modelling, particularly during MUMBA (the lead author has three other publications on the isoprene bias, cited in this manuscript). We have made the std deviation plot as requested (below), although the colour of the CC_γT+LEF run was changed to red to show up better. There is wide variation in observed isoprene, particularly during SPS1. The model also shows high variation before dawn in SPS1 and MUMBA. We calculate the percentage of modelled hours which are within +1 standard deviation of the observed isoprene as follows:

Base run (CC_γT+LEF run). Bringelly = 40% (90%); SPS1 = 89% (100%); MUMBA = 19% (33%).

[Figure]

The point of this paper was to see if we could use the new measurements to reduce the modelled biases through the sensitivity runs, which we have done. The impacts of the future simulations are then explored via their differences on the least biased current climate run, not absolute values. I think we've made the point that the modelled isoprene was never perfect to start with, and much additional discussion of the biases is not necessary. We will include the figure in the supplementary.

At line 267 "The average modelled isoprene in the CC_$\gamma$T+LEF run is within +/- 1 standard deviation of the observations 90 – 100 % of the time during Bringelly and SPS1, and 33 % during MUMBA which continues to exhibit high bias (supplementary figure S4)".

*In which year was air quality in Sydney and Darwin classed as 'very good' 2019? Or in many years?*
This relates to the latest Australian State of the Environment reporting period of 2009-2014.
Line 309 "(years 2009 – 2014)"

*Improving the LEF did more to decrease the bias than the temperature response measurements? I think this is an important findings. Looking at the model vs observations, improving the LEF did more to decrease bias than the improved temperature response - i.e. dark blue line compared obs vs light blue line compare to obs.*
We knew that decreasing the LEF by about 40% would bring the magnitude of the model into better agreement with the observations from our previous work (see above). However, the largest increase in the $r^2$ fit of the data actually occurred when using the temperature response data alone (for Bringelly and MUMBA) than the decrease in LEF. Correctly fitting the SPS1 observed data is more difficult due to the spike at 10am.

*Would peak in temperature response be different if the measurements were conducted on older leaves? Would it be expected that the peak temp in the temp response also be different? Or is it just the total emissions? As I mentioned earlier, I would recommend that this issue about new and old leaves and the impact here is unpacked a bit more.*
Without further experimentation using older trees it is impossible to answer the reviewer's question and any comment would be pure speculation. However, in relation to the reviewer's earlier comment, we have added some more discussion at line 347.

"Our measurements were conducted on sapling trees which may exhibit higher isoprene emissions than adult trees when emission rates are expressed on leaf mass basis but not on a leaf area basis (Street et al., 1997). Street et al. (1997) explained this through younger leaves having a higher specific leaf area than older leaves because eucalypts exhibit heterophylly (the foliage leaves on the same plant are of two distinctly different types). The apparent difference in emission rates between young and old leaves could be a consequence of morphology rather than biochemistry, so we expect the trend between the current and future climate growth emissions to be similar amongst trees of all ages."

*Similar to a comment above - which eucalyptus were used in MEGAN to estimate these characteristics?*
See comment above

**References**

Emmerson, K. M., Palmer, P. I., Thatcher, M., Haverd, V. and Guenther, A. B.: Sensitivity of isoprene emissions to drought over south-eastern Australia: Integrating models and satellite observations of soil moisture, Atmospheric Environment, 209, 112–124, doi:10.1016/j.atmosenv.2019.04.038, 2019.

Guenther, A. B., Zimmerman, P. R., Harley, P. C., Monson, R. K. and Fall, R.: Isoprene and monoterpene emission rate variability: Model evaluations and sensitivity analyses, Journal of Geophysical Research: Atmospheres, 98(D7), 12609–12617, doi:10.1029/93JD00527, 1993.

Street, R. A., Hewitt, C. N. and Mennicken, S.: Isoprene and monoterpene emissions from a eucalyptus plantation in Portugal, Journal of Geophysical Research: Atmospheres, 102(D13), 15875–15887, doi:10.1029/97JD00010, 1997.